# Impact of Perceived Social Support on the Relationship between ADHD and Depressive Symptoms among First Year Medical Students: A Structural Equation Model Approach

**DOI:** 10.3390/children8050401

**Published:** 2021-05-16

**Authors:** Nuntaporn Karawekpanyawong, Tinakon Wongpakaran, Nahathai Wongpakaran, Chiraphat Boonnag, Sirinut Siritikul, Sirikorn Chalanunt, Pimolpun Kuntawong

**Affiliations:** 1Department of Psychiatry, Faculty of Medicine, Chiang Mai University, Chiang Mai 50200, Thailand; nuntaporn.karawek@cmu.ac.th (N.K.); nahathai.wongpakaran@cmu.ac.th (N.W.); pimolpunssj@gmail.com (P.K.); 2Faculty of Medicine, Chiang Mai University, Chiang Mai 50200, Thailand; Chiraphat_b@cmu.ac.th (C.B.); sirinut_siritikul@cmu.ac.th (S.S.); sirikorn.chalanunt@cmu.ac.th (S.C.)

**Keywords:** adult ADHD, undergraduate, family, friends, depression

## Abstract

Background: Attention deficit hyperactivity disorder (ADHD) is associated with depression among college students, while perceived social support is also associated with depression, especially among young adults. This study aimed to examine to what extent perceived social support mediated the relationship between ADHD symptoms and depressive symptoms. Methods: In total, 124 first year medical students completed the Adult ADHD Self-Report Scale Screener (ASRS), the Patient Health questionnaire-9 and the revised Thai Multidimensional scale of perceived social support reflecting ADHD symptoms, depressive symptoms, and perceive social support, i.e., family members, friends and other significant people, respectively. Structural equation modeling was used to investigate the hypothesized mediation model. Results: ADHD symptoms exhibited a significant indirect effect on depressive symptoms via perceived social support. ADHD symptoms initially had a direct effect on depression; thereafter, it reduced to a non-significance effect after perceived social support was added. The total variance explained by this model was 35.2%. The mediation model with family support as a mediator showed the highest effect size. Conclusions: The study highlighted the importance of perceived social support, particularly family support, on depressive symptoms among young medical students experiencing ADHD symptoms. The model suggests promising relationships for further research on ADHD-related depression and potential treatment in the future.

## 1. Introduction

Attention deficit/hyperactivity disorder (ADHD) is a neuropsychiatric condition affecting preschoolers, children, adolescents, and adults worldwide. ADHD is characterized by a pattern of diminished sustained attention and increased impulsivity or hyperactivity. Patients with a diagnosis of ADHD might have trouble finishing tasks, sitting still or keeping track of things, appointments or details [1,2]. The prevalence of ADHD is 5.29% and 4.4% among children and adults in the general population, respectively [3,4]. The prevalence of ADHD among college students is 7.6% [5]. The range of ADHD prevalence among medical students was reported as 5.5 to 23.7%, depending on culture and measurement [6,7,8]. No definite prevalence of ADHD has been reported in Thailand, including among Thai medical students. As the literature shows, the worldwide prevalence of ADHD in various populations is about 5% [9]. We suggest that ADHD problems among Thai medical students exist even though no report has confirmed this as of yet. The existence of ADHD symptoms at present may suggest the presence of childhood ADHD or underdiagnosed childhood ADHD. Although diagnostic interviews have not been conducted, sometimes depressive symptoms in young adults with ADHD can be related to untreated baseline ADHD, which undermines their academic progression or requires them to make an extra effort to reach a similar performance level compared with their peers without ADHD [10]. Research has revealed that 27.2% of medical students experience depression, 11% of whom experience suicidal ideation [11]. The authors would therefore like to focus on examining depression related to ADHD symptoms, even though the definite rate of prevalence of ADHD is unknown.

The association between ADHD and depression has been well-documented [12,13]. Having ADHD puts medical students at seven times the risk of depression [8]. The presence of ADHD symptoms with depressive symptoms highlights the need to understand factors that may account for their high co-occurrence. What contributes to the relationship between ADHD symptoms and depression is elucidated in a number of hypotheses. For example, it might be related to self-perceived competencies. According to Cole, self-perceived competencies were significantly predicted by appraisals from teachers, parents and peers in the developmentally essential domains, including academic competence, social acceptance, physical appearance, behavioral conduct and sports competence. Self-perceived competencies were negatively related to later levels of self-reported depressive symptoms [14,15]. Medical students diagnosed with ADHD may also increase their chances of becoming depressed because of academic performance deficits. They may find it difficult to maintain satisfactory academic performance and suitable work performance due to inattention and higher-order executive function deficits. Studies show children with attention problems tend to have difficulty completing assignments and meeting teachers’ expectations; for these reasons, they might face many adverse outcomes, including learning problems, teacher conflict and school dissatisfaction [16,17,18,19,20].

Another hypothesis is related to social rejection; significant covariates for depressed mood resulted from disruptive behavior problems [21,22]. Children diagnosed with ADHD scored lower on social preference, higher on social impact, were less well-liked and more often in the rejected social status category [23].

Moreover, the parent–child relationship may mediate depression risk among children with a diagnosis of ADHD. Related studies have revealed that those children may perceive both parent–child rejection and hostility due to parental cognitions about child behaviors and parenting stress. Children with a diagnosis of ADHD require tremendous parental efforts, which may disrupt normative parenting behaviors. Parent–child relationships were vulnerable to disruption, and these parent–child problems increased the risk of depression [24,25]. In addition to the parent–child problem, perceived low parental support (autonomy, involvement and warmth) was associated with depression among emerging adults with a diagnosis of ADHD [26]. Meinzer and colleagues have demonstrated that the relationship with the mother but not friendship mediated the relationship between ADHD symptoms and depressive symptoms among adolescents [27].

Social support is a well-known protective factor for depression in a variety of populations and clinical settings [28,29,30,31]. Emerging adulthood is the developmental period of medical students. This period represents a time of semi-autonomy where individuals begin to take on some responsibilities of independent living. Sources of support for such medical students include more than their parents, such as peers and other special or significant people including beloved individuals and teachers [32]. Supportive behaviors can come in varying forms, including emotional support, e.g., acceptance and caring, providing information or advice, providing feedback and providing needed resources, e.g., time, money and help.

The aforementioned evidence as well as related studies have revealed the association of perceived social support, mostly from the parents among children with a diagnosis of ADHD. Long-term studies in children with a diagnosis of ADHD have demonstrated that ADHD symptoms remain over time, at least into the emerging adult period [33,34]. One study showed that 65% of individuals with an ADHD diagnosis had partial remission at the age of 25 [35]. Recent study has paid more attention to increasing our understanding of the continuity of ADHD symptoms in young adults [36]; in particular, the DSM criteria allow diagnosis of ADHD symptoms to be provided across the lifespan and not only during childhood [37]. The present study focused on ADHD symptoms reported among first year medical students who were considered late adolescents. In addition, little knowledge is available regarding the influence of perceived social support among emerging adults with ADHD symptoms experiencing depression, as well as the effect of perceived social support from sources other than parents. This study aimed to examine the extent to which the different domains of perceived social support, i.e., family members, friends and other significant people, mediated the relationship between ADHD symptoms and depressive symptoms in a group of emerging adult first year medical students. Particularly, we would like to explore to what extent perceived family support came into effect, and how the effect of support from friends and other significant people would play out. We hypothesize that overall perceived social support would reduce the strength of the relationship between ADHD symptoms and depressive symptoms. Based on the collectivistic culture of the sample, we also hypothesize that among three sources of social support, perceived support from family would remain a strong influence on the relationship between ADHD symptoms and depressive symptoms among these Thai adolescents.

## 2. Materials and Methods

### 2.1. Participants and Procedures

This survey was conducted among 124 first year medical students in Chiang Mai University, Thailand in 2017. Each participant provided written informed consent before completing the questionnaires, which included sociodemographic data and records related to parents. Inclusion criteria for eligible students were (1) age range from 16 to 21 years old; and (2) first year medical students. The only exclusion criterion was a refusal of the informed consent. They also completed questionnaires concerning Adult ADHD, perceived social support, and depressive symptoms. Ethics approval was obtained from the Faculty of Medicine, Chiang Mai University, Thailand, before taking any further steps in the research (see also “Institutional Review Board Statement” below for more information).

### 2.2. Instruments

#### 2.2.1. General Demographic Data

The self-reported questionnaires consisted of sociodemographic and health-related characteristics including sex, age, and underlying diseases including mental health difficulties. Moreover, information on the parents’ education and parents’ occupation was collected.

#### 2.2.2. Adult ADHD Self-Report Scale (ASRS) Screener V1.1

ASRS, developed by Kessler et al., is a six-item self-rated questionnaire evaluating ADHD symptoms using the DSM-IV-TR among adults. Among six items, four items measure inattention (items 1–4), and two measure hyperactivity (items 5–6; [38]. These six items are (1) trouble wrapping up the final details of a project, (2) difficulty getting things in order, (3) problems remembering appointments or obligations, (4) avoiding or delaying getting started, (5) fidgeting or squirming with your hands or feet when sitting for a long time, and (6) feeling overly active and compelled to do things. The responses range from 0 to 4 (never to very often). Participants with higher total scores, have higher levels of ADHD symptoms. When an individual has at least four symptoms, they are classified as having ADHD symptoms. The Thai version of the ASRS-v1.1 screener was validated and used to screen adults with ADHD, and it showed a sensitivity of 0.93 and a specificity of 0.71 [39,40]. The Cronbach’s alpha for the study sample was 0.80.

#### 2.2.3. Patient Health Questionnaire (PHQ)-9

The PHQ-9 is a 9-item self-reporting questionnaire measuring the extent to which an individual has experienced depressive symptoms over the past two weeks [41]. The four-response Likert scale ranges from 0 (not at all) to 3 (nearly every day). Each question results in a score from 0 to 3, all of which are combined into a final score. The higher the total score, the higher the level of depressive symptoms. The score ranges from 0 to 27 and can be divided into five levels of depressive severity: 0–4, none; 5–9, mild; 10–14, moderate; 15–19, moderately severe; 20–27, severe. The Thai version PHQ-9 showed a Cronbach’s alpha of 0.79 and a positive association between the PHQ-9 and the HAM-D (r = 0.56, *p* < 0.001; [42]. The study sample showed a Cronbach’s alpha of 0.85.

#### 2.2.4. Revised-Thai Multidimensional Scale of Perceived Social Support (r-MSPSS)

The r-MSPSS is a 12-item self-reporting questionnaire measuring the extent to which an individual has experienced being supported by significant others, friends, and family members [32]. The 7-response Likert scale ranges from 1 (strongly disagree) to 7 (strongly agree). The higher the total score, the higher the level of perception of being supported. The revised Thai version MSPSS showed a Cronbach’s alpha of 0.79 and supported a three-factor solution model [43]. The study sample showed a Cronbach’s alpha of 0.85.

### 2.3. Data Analysis

Descriptive analysis was used for sociodemographic data. The frequency, percentage and mean standard deviation of ADHD symptoms, r-MSPSS and PHQ-9 scores were reported. Before building the mediation model, the relationship between ASRS, MSPSS and PHQ-9 scores was examined using Pearson’s correlation coefficient to ensure a significant relationship among variables.

To test for the mediating effect of perceived social support, a mediation model can be created either by a simple mediation model (observed variables are used) or a mediation model within the structural equation modeling (SEM) framework. The advantage of SEM over the path model is that SEM reduces random measurement errors, which is absent using the standard regression method [44]. In this model, ASRS served as the independent variable (X), MSPSS as the mediator (M), and PHQ-9 as the dependent/outcome variables (Y) (Figure 1). To ensure that SEM can be adopted in such a sample size, item reduction using a parceling method was applied. Based on the following determined input: anticipated effect size, 0.3; desired statistical power, 0.8; number of latent variables, 3; number of observed variables, 13–14 (including the covariates of age and sex). Therefore, four parceling observed variables were used for each latent factor, with a probability level of 0.05; it yielded a sample size of 119 [45].

Before SEM analysis, data were examined for skewness, kurtosis, and multivariate normality. All variables were shown to have skewness and kurtosis within the acceptable range of <±3 [46]; the ordered values of N were non-significantly far from the centroid, indicating no violation of multivariate normality was observed. Minor missing data were rectified by series mean, and standardized parameter estimates were calculated. Indirect effects were tested using bootstrap estimation with 10,000 samples and percentile bootstrap confidence intervals were reported at the 95% confidence level [44]. The null hypothesis was rejected, meaning that it significantly differed from zero. To put it another way, the direct/indirect effect was considered significant when the 95% confidence interval for the standardized estimated coefficient of the direct/indirect effect did not include zero.

Based on the following determined input, anticipated effect size, 0.3; desired statistical power, 0.8; number of latent variables, 3; number of observed variables, 13–14 (including the covariates of age and sex). The probability level was 0.05, which yielded a sample size of 119 [45].

MSPSS divided the sources of support into three sources. Referring to overall social support may not give sufficient information because each source may have a different impact at different stages of life. In particular, to children and adolescents, family support is important, while adults may find other significant people such as doctors and bosses important as well [47]. We tested four models (M1 to M4) based on MSPSS (Figure 1 and Figure 2). Model 1’s latent variable represented overall perceived social support and three indicators representing family members, friends and other significant people; Model 2’s latent variable represented perceived family members’ support and three indicators representing four items assessing family support; Model 3’s latent variable represented perceived friends’ support and four indicators representing four items assessing friends’ support; Model 4’s latent variable represented perceived other significant people’s support and four indicators representing four items assessing other significant people’s support. Each measurement model was tested and exhibited acceptable fit statistics before running SEM.

For each latent construct, four manifest indicators were considered (except Model 1). The items of the ADHD and PHQ-9 questionnaire were reduced to four parcels. The parcels were constructed according to Little et al. [48], whereupon those items with the highest factor loadings of the latent variable were allocated alternately to both parcels in descending order.

To test for model fitness, the following model fit indices were employed: the minimum ratio of chi-square divided by its degrees of freedom (χ^2^/DF); the root mean square error of approximation (RMSEA); the comparative-fit index (CFI); and the Tucker–Lewis Index (TLI). For a good model fit, the ratio χ^2^/DF should be <3, and CFI and TLI should be higher than 0.95 [49], whereas values greater than 0.90 are usually interpreted as indicators for an acceptable fit. Moreover, RMSEA values < 0.06 indicate a good model fit, and values of 0.08 still reflect an adequate fit. The model was tested using a maximum likelihood estimation method for covariance matrices. For all analyses, the level of significance was *p* < 0.05. All statistical analyses were performed using the program IBM SPSS, 22.0 and AMOS, Version 18 (IBM Corp, Armonk, NY, USA).

## 3. Results

Of 124 participants, most were female, with a mean age of 18.17 ± 0.7 years. One-third of the participants reported having allergies as an underlying disease. None reported having any mental health or psychiatric illnesses, including history of ADHD. Slightly over half of the participants (52.4%) were firstborn. A quarter of participants had ADHD symptoms based on the ASRS screener criteria. No significant difference between the ADHD symptoms and non-ADHD symptoms groups was observed, except for the number of siblings (*p* < 0.05). Other details are shown in Table 1.

Table 2 Correlation matrix among the variables, indicating all were significantly correlated except for ASRS scores and MSPSS-FR.

Initially, the latent variable of ASRS had a significantly direct effect on the latent variables of PHQ-9 (β = 0.311, 95% CI 0.116, 0.520, *p* = 0.008), which then reduced (=0.145, 95%CI −0.079, 0.340, *p* = 0.292) after MSPSS was added to the SEM. Table 3 shows the path coefficients denoting direct and indirect effects of the ASRS latent variable on PHQ-9 latent variables. Model 1 shows the significant indirect effect of the latent variables of ASRS through overall perceived social support (β = 0.168 (95% CI 0.039, 0.367)); the variance explained by this model was 35.2%. The fit statistics of Model 1 were as follows: Chi-square = 67.963, degrees of freedom = 57, *p* = 0.152, Chi-square/df = 1.19. CFI = 0.971, TLI = 0.960, and RMSEA = 0.040 (90%CI = 0.000–0.071).

For Model 2, the significant indirect effect of the latent variable of ASRS through overall perceived family members’ support (β = 0.116 (95% CI 0.037–0.256)); the variance explained by this model was 24.4%. The fit statistics of Model 1 were as follows: Chi-square = 74.352, degrees of freedom = 69, *p* = 0.308, Chi-square/df = 1.08. CFI = 0.991, TLI = 0.988 and RMSEA = 0.025(90%CI = 0.000–0.060).

For Model 3, the non-significant indirect effect of the latent variable of ASRS through overall perceived friends’ support (β = 0.068 (95% CI 0.000–0.084)); the variance explained by this model was 25.3%. The fit statistics of Model 1 were as follows: Chi-square = 83.619, degrees of freedom = 69, *p* = 0.111, Chi-square/df = 1.21. CFI = 0.977, TLI = 0.969 and RM = 0.042 (90%CI = 0.000–0.070).

For Model 4, the non-significant indirect effect of the latent variable of ASRS through overall perceived significant people’s support (β = 0.050 (95% CI 0.000–0.110)); the variance explained by this model was 19.8%. The fit statistics of Model 1 were as follows: Chi-square = 114.02, degrees of freedom = 68, *p* < 0.001, Chi-square/df = 1.67. CFI = 0.931, TLI = 0.908 and RM = 0.074 (90%CI = 0.049–0.097).

## 4. Discussion

This study aimed to examine whether perceived social support was able to mediate the relationship between ADHD symptoms and depressive symptoms among young adults, i.e., first year medical students, and the results supported our hypotheses. Notably, perceived family support was the only significant mediator compared with the rest, highlighting the importance of perceived family support among these emerging adults. Other sources of support existed even though they were non-significant. These findings support related studies conducted by Meinzer and colleagues regarding the significant role of parents [26]. Their recent study showed that the negative relationship between a mother and adolescent remained a predictor for developing depression in a longitudinal fashion, while a negative friendship did not [27], which is consistent with the present study but in a different age group. In the study by Meinzer and colleagues, participants were adolescents, while in the present study, they were emerging adults.

In fact, in older age participants such as in the present study, other sources of support, especially friends, should come into play, but this was not the case. The reason perceived as to why family support prevailed, despite the participants being emerging adults, may have been the fact that the participants live in a collectivistic society and are influenced by Thai family tradition; this influence is present in many cultures, especially in Asian countries [50,51,52]. Compared with medical students from Western cultures, the role of the family is distinctively predominant [53]. It could be interpreted positively as ‘emotional support’ or negatively as ‘overinvolvement’ depending on many factors.

Based on related studies, however, perceived family support was vital for Thai medical students, especially first year students, implying that they maintained strong ties with their families [54,55]. Being supported by family is important not only to help them grow psychologically and achieve academically, but also it could help students with ADHD symptoms reduce the tendency to develop psychological distress when they experience serious conflicts and more adverse life events [56]. Additionally, this support could help when encountering external factors, i.e., academic stressors derived within the medical school environment [57]. We do not fully understand the mechanism of how family support was associated with decreased depressive symptoms. Other factors related to ADHD symptoms such as self-esteem and loneliness might be mitigated when they receive support.

Moreover, based on psychological development, the significance of perceptions of friends and other significant people should be more pronounced in later years of study. Whether those supports have a sufficient effect to mediate the relationship between ADHD symptoms and depressive symptoms as does family support remains unknown, and this should be a subject of further investigation.

A concern that should be raised here is what action to take when some students reported high scores on ADHD or depressive symptoms. No doubt these symptoms impact students’ wellbeing as well as their academic performance. The researcher is normally not allowed to provide direct interpretation according to the participant’s confidentiality (except for suicidality). However, this concern is mitigated as students are usually informed about consultation services when they are made aware of having problems, instigated by those questionnaires.

### Strengths and Limitations

Perceived social support being associated with decreased depressive symptoms is possibly an innovative finding of this study. The strength of the present study is that it provides some insight about the different impacts of each source of social support.

Some limitations should be noted here. (1) The sample size was relatively small, even though it met the minimal requirements for conducting SEM, and (2) it included only first year medical students. Further replication studies either within or outside medical students are strongly encouraged. (3) Some items of the ASRS scale could give a false-positive result, e.g., item 6 (high level of energy). If this self-report is to be used again, other psychiatric inventories should be added to help discriminate symptoms, e.g., hypomanic symptom. Notably, the study was conducted using the DSM-IV-TR because it represents the only validated version. However, the new ASRS screener using DSM-5 appears to have similar items. (4) This study did not have data to rule out the effect of psychoactive substance use, which can mimic inattention and hyperactivity symptoms. In addition, other related factors left unstudied include sleep problems. These issues should also be explored in further studies. (5) We cannot conclude any causal relationship from this cross-sectional design, so a longitudinal study in this population should provide more robust evidence in that regard.

Lastly, this study represents only the relationship of ADHD symptoms and depressive symptoms, without definite diagnosis of ADHD and depressive disorders; we cannot confidently determine the magnitude of the mediated effect. Moreover, a further study should be conducted to examine the scale of ADHD, comorbid depressive disorders and its relationship among adolescents as well as Thai medical students.

## 5. Conclusions

To the best of our knowledge, the present study has emphasized the importance of perceived family support among young adults assumed to be carried over from adolescence, as demonstrated before by related studies. The present study also adds some findings related to sources of support other than family members. This may be one of the earliest studies among medical students experiencing ADHD symptoms. Interestingly, however, exploring to what extent the ADHD symptoms would change and how the three sources of support would impact the relationship between ADHD symptoms and depressive symptoms, especially in later years in medical education in the longer term, is encouraged.

## Figures and Tables

**Figure 1 children-08-00401-f001:**
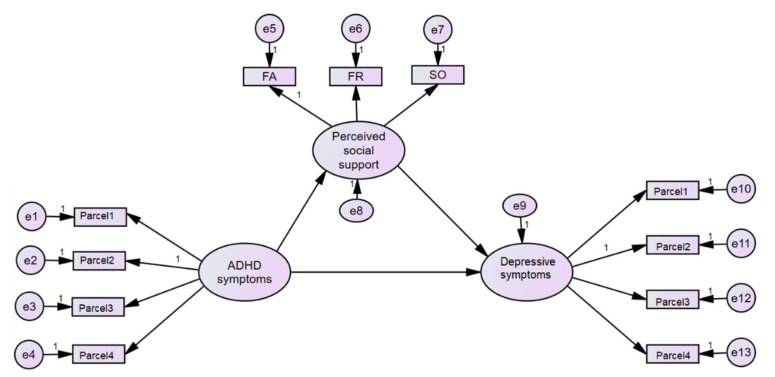
Mediation model analyses using structural equation model diagram—overall perceived social support (Model 1). Note: ASRS—Adult ADHD Self-Report Scale; rMSPSS—revised Thai Multidimensional Scale of Perceived Social Support; SO—significant others; FA—family; FR—friends; PHQ—Physical health questionnaire; e—error term.

**Figure 2 children-08-00401-f002:**
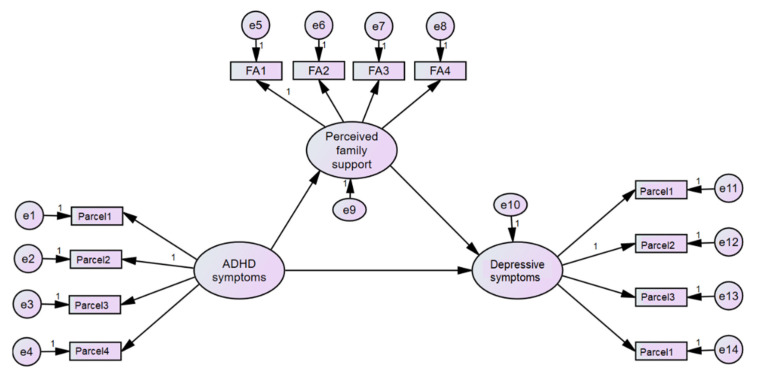
Mediation model analyses using structural equation model diagram—perceived family support (Model 2). Note: ASRS—Adult ADHD Self-Report Scale; rMSPSS—revised Thai Multidimensional Scale of Perceived Social Support; SO—significant others; FA—family; FR—friends; PHQ—Physical health questionnaire; e—error term.

**Table 1 children-08-00401-t001:** Sociodemographic and clinical characteristics.

	All (n = 124)	ADHD Symptoms (n = 31)	Non-ADHD Symptoms (n = 93)	*p*-Value
Variable	n (%) or Mean ± SD	n (%) or Mean ± SD	n (%) or Mean ± SD	
Male sex, n (%)	46 (37.1)	10 (32.3)	36 (38.7)	0.520
Age (years old), mean ± SD	18.78 ± 0.74	18.55 ± 0.64	18.75 ± 0.77	0.186
Underlying illness, n (%)				
Allergy	39 (31.5)	8 (25.8)	23 (24.7)	0.905
Mental health problem or psychiatric disorder	0 (0)	0 (0)	0 (0)	-
Father’s years of education, SD Mean ± SD	14.98 ± 5.6	14.85 ± 6.0	15.03 ± 5.4	0.887
Father’s mental illness, n (%)	1(0.8)	1(3.3)	0(0)	0.080
Mother’s years of education, Mean ± SD	14.71 ± 5.7	15.46 ± 5.8	14.45 ± 5.6	0.437
Mother’s mental illness, n (%)	0(0)	0(0)	0(0)	-
Single child, n (%)	18 (14.5)	2 (6.5)	16 (17.2)	0.141
Firstborn, n (%)	65 (52.4)	14 (45.2)	51 (54.8)	0.350
Lastborn, n (%)	46 (37.4)	11 (36.7)	35 (37.6)	0.924
Number of siblings, Mean ± SD	2.1 ± 0.79	2.37 ± 0.85 *	2.01 ± 0.76	0.036
ASRS total score	10.11 ± 3.54	14.45 ± 1.8	8.67 ± 2.7	<0.0001
rMSPSS-total	5.49 ± 0.9	5.38 ± 0.9	5.53 ± 0.9	0.458
rMSPSS-SO	4.98 ± 1.4	4.92 ± 1.6	5.00 ± 1.4	0.800
rMSPSS-FA	6.07 ± 1.0	5.82 ±1.2	6.15 ± 0.9	0.117
rMSPSS-FR	5.43 ± 1.0	5.41 ± 0.9	5.43 ± 1.0	0.931
PHQ-9	7.19± 3.86	7.87 ± 4.0	6.97 ± 3.8	0.261

Notes: Abbreviations: * significant difference between two groups; SD, standard deviation; ADHD, Attention deficit disorder and hyperactivity disorder; ASRS, Adult ADHD Self-Report Scale; rMSPSS, revised Thai Multidimensional Scale of Perceived Social Support; SO, significant others; FA, family; FR, friends; PHQ, Physical health questionnaire.

**Table 2 children-08-00401-t002:** Correlation between variables.

Variable	ASRS	rMSPSS-SO	rMSPSS-FR	rMSPSS-FA	PHQ-9
ASRS	−				
rMSPSS−SO	−0.183 *	−			
rMSPSS−FR	−0.164	0.469 **	−		
rMSPSS−FA	−0.238 **	0.335 **	0.338 **	−	
PHQ−9	0.264 **	−0.199 *	−0.320 **	0.368 **	−

Notes: Abbreviations: * *p* < 0.05; ** *p* < 0.01; ASRS, Adult ADHD Self-Report Scale; rMSPSS, revised Thai Multidimensional Scale of Perceived Social Support; SO, significant others; FA, family; FR, friends; PHQ, Physical health questionnaire.

**Table 3 children-08-00401-t003:** Direct and indirect effects of ASRS on PHQ-9 controlling for sex and age.

Variable	Coefficient	S.E.	LLCI	ULCI	*p*-Value	R Square
Model 1. Mediator: Overall Perceived Social Support						
Total effect	0.313	0.119	0.109	0.505	0.011	0.352
Total indirect	0.168	0.103	0.039	0.367	0.029	
Direct effect	0.145	0.128	−0.079	0.340	0.292	
Model 2. Mediator: Perceived Family Members’ Support						
Total effect	0.312	0.119	0.103	0.499	0.012	0.244
Total indirect	0.116	0.063	0.037	0.256	0.008	
Direct effect	0.196	0.111	0.030	0.381	0.044	
Model 3. Mediator: Perceived Friends’ Support						
Total effect	0.312	0.119	0.106	0.504	0.012	0.253
Total indirect	0.068	0.046	0.000	0.084	0.102	
Direct effect	0.243	0.120	0.021	0.259	0.045	
Model 4. Mediator: Perceived Significant Others’ Support						
Total effect	0.310	0.119	0.137	0.475	0.003	0.198
Total indirect	0.050	0.032	0.000	0.110	0.085	
Direct effect	0.261	0.104	0.087	0.430	0.013	

Notes: Abbreviations: ASRS, Adult ADHD Self-Report Scale; PHQ, Physical health questionnaire; S.E., standard error; LLCI, lower level of confidence interval; ULCI, upper level of confidence interval.

## Data Availability

The dataset is available upon reasonable request to Tinakon Wongpakaran (email: tinakon.w@cmu.ac.th).

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
