# Peer review of "Impact of Perceived Social Support on the Relationship between ADHD and Depressive Symptoms among First Year Medical Students: A Structural Equation Model Approach"

_children, 2021, doi:10.3390/children8050401_

Round 1

Reviewer 1 Report

The use of mathematical modeling in explaining biological phenomena is a valuable tool. Unfortunately, the presented work has several shortcomings.

The purpose of the study is not clear because the survey was conducted on a group of first-year medical students generally rather than on students with a diagnosis of ADHD. I am not entirely convinced that ADHD is more common among medical students - even 23%. ADHD is a disease with attention deficit hyperactivity disorder, which may be disqualifying in undertaking such demanding studies.
Therefore, in the first stage of the study, the scale of the problem should be assessed, i.e., the incidence of ADHD, depression, and then comorbid depression in the course of ADHD among medical students in the Taiwanese population.

It is also nothing new that the support of the family and loved ones are an essential protective factor not only for depression but also for other mental disorders (PMID: 27445355, PMID: 28695369, PMID: 29843538)

The study did not exclude people using/abusing psychoactive substances, which may significantly affect the results of the self-report questionnaires.

Since ASRS is a questionnaire for screening adult ADHD, the natural stage of the analysis should be separating two groups of students adequately to "When an individual has at least four symptoms, they are classified as having ADHD symptoms."

Were students with 4 ADHD symptoms referred to a specialist?

In the case of PHQ-9, the description: "The higher the total score, the higher the level of depressive symptoms" is insufficient. Ranges for symptom severity or at least dichotomous differentiation of presence/absence of depressive symptoms should be given.

1/4 of the subjects had ADHD symptoms, and 1/3 had allergies. Since "Early food allergy and respiratory allergy symptoms independently and synergistically contributed to a higher risk of ADHD (PMID: 29524252)," one has to analyze the presence of allergy as a correction variable.

Although there were slightly more girls in the study, the gender distribution among people with ADHD symptoms was not shown, especially since the male gender is more often affected.

The choice of four M1-M4 models requires a more detailed justification.

How do the authors define the direct/indirect influence of social support on the occurrence of depression in students with ADHD symptoms? Is this approach justified only for the studied group, or can it be considered a mechanism of general importance for the population?

Author Response

Dear Editor and reviewers,

We greatly appreciate your time for your kind review and constructive comments on the manuscript entitled “Impact of perceived social support on the relationship between ADHD and depressive symptoms among first year medical students: a structural equation model approach”

We have carefully studied your comments and made corrections as suggested. Please find enclosed the revised version of our manuscript high-lighted in green and the detailed point-by-point response to the comments raised by the reviewers.

We hope that our revised manuscript will be satisfactorily acceptable.

We are looking forward to hearing from you.

Best regards,

Tinakon Wongpakaran and colleagues

Reviewer 2

The authors are attributing depression symptoms to ADHD, yet, according to DSM criteria, to make a diagnosis of ADHD: * "The symptoms do not happen only during the course of schizophrenia or another psychotic disorder. The symptoms are not better explained by another mental disorder (e.g. Mood Disorder, Anxiety Disorder, Dissociative Disorder, or a Personality Disorder)." It is much more likely that Depression accounts for the ADH type symptoms of restlessness, inattentiveness, etc. than the other way around. There is also literature that inadequate sleep can result in ADH type symptoms and the rigors of medical school may well result in inadequate sleep. The paper goes into some detail on the statistical and analytic methods used for their models, for which I do not have the expertise to comment.

Response: Thank you for this comment. You are right about the diagnosis of ADHD based on DSM criteria. However, in this study, we cannot be certain whether or not the individuals in the sample have ADHD or depressive disorders because that requires clinicians to perform a diagnosis. On the contrary, in this study, we rely on self-reporting, and therefore only “symptoms” of ADHD and depression can be claimed.

In addition, we agree with the reviewer that sleep may play a role in ADHD symptoms, especially in medical students. However, we did not collect these data. We add this issue to the discussion of the study’s limitations.

In addition, other related factors left unstudied include sleep problems, especially in medical students. This issue should also be explored in further studies.

Reviewer 1

The use of mathematical modeling in explaining biological phenomena is a valuable tool. Unfortunately, the presented work has several shortcomings.

The purpose of the study is not clear because the survey was conducted on a group of first-year medical students generally rather than on students with a diagnosis of ADHD. I am not entirely convinced that ADHD is more common among medical students - even 23%. ADHD is a disease with attention deficit hyperactivity disorder, which may be disqualifying in undertaking such demanding studies.
Therefore, in the first stage of the study, the scale of the problem should be assessed, i.e., the incidence of ADHD, depression, and then comorbid depression in the course of ADHD among medical students in the Taiwanese population.

Response: Thank you for your suggestion. We have revised our manuscript as follows.

No definite prevalence of ADHD has been reported in Thailand, including among Thai medical students. As the literature shows, the worldwide prevalence of ADHD in various populations is about 5%[1]. We believe that ADHD problems among Thai medical students exist even though no report has confirmed this as yet. The existence of ADHD symptoms at present may suggest the presence of childhood ADHD or underdiagnosed childhood ADHD. Despite the fact that we cannot accurately estimate the magnitude of the ADHD problem, the fact that ADHD symptoms interfere with psychological wellbeing is important. Research has revealed that 27.2% of medical students experience depression, 11% of whom suffered from suicidal ideation[2]; the authors would therefore like to focus on examining depression related to ADHD symptoms.

It is also nothing new that the support of the family and loved ones are an essential protective factor not only for depression but also for other mental disorders (PMID: 27445355, PMID: 28695369, PMID: 29843538)

Response: Thank you for providing us with more references, especially regarding children. This makes it clearer that social support is related to depression, however, when it comes to three or more variables. We found in some literature that social support had less impact on depression when compared to some other variables such as perceived stress[5]. In addition, even though social support is related to depression, its relationship to ADHD symptoms cannot be ascertained.

The study did not exclude people using/abusing psychoactive substances, which may significantly affect the results of the self-report questionnaires.

Response: Thank you for raising this point; we agree. We did not add a direct question addressing this issue but rather recruited general information on psychiatric disorders, which might overlook psychoactive substances. We have added this issue to the discussion of the study’s limitations as follows:

Limitations: This study did not have the data to rule out the effect of psychoactive substance use, which can mimic inattention and hyperactivity symptoms. This issue should be explored in further study.

Since ASRS is a questionnaire for screening adult ADHD, the natural stage of the analysis should be separating two groups of students adequately to "When an individual has at least four symptoms, they are classified as having ADHD symptoms."

Response: We thank you for your suggestion. We have added more analyses comparing the two groups in Table 1 as suggested. In addition, some corrections have been made regarding the values of rMSPSS.

Were students with 4 ADHD symptoms referred to a specialist?

Response: Following ethical regulations, only those who have serious psychological symptoms, that is, suicidal ideation, are contacted by the research team and advised to seek consultation for this serious psychological condition.

In the case of PHQ-9, the description: "The higher the total score, the higher the level of depressive symptoms" is insufficient. Ranges for symptom severity or at least dichotomous differentiation of presence/absence of depressive symptoms should be given.

Response: We have revised as follows: 

Each question results in a score from 0 to 3, all of which are combined into a final score. The higher the total score, the higher the level of depressive symptoms. The score ranges from 0 to 27 and can be divided into five levels of depressive severity: 0–4, none; 5–9, mild; 10–14, moderate; 15–19, moderately severe; 20–27, severe.

1/4 of the subjects had ADHD symptoms, and 1/3 had allergies. Since "Early food allergy and respiratory allergy symptoms independently and synergistically contributed to a higher risk of ADHD (PMID: 29524252)," one has to analyze the presence of allergy as a correction variable.

Response: Thank you for your keen observation and citation. We have examined the difference in allergies between those with ADHD symptoms and those without ADHD symptoms but it was nonsignificant in this sample (see revised Table 1).

Although there were slightly more girls in the study, the gender distribution among people with ADHD symptoms was not shown, especially since the male gender is more often affected.

Response: New results regarding gender distribution have been added in Table 1. No difference between ADHD and non-ADHD groups has been noted.

The choice of four M1-M4 models requires a more detailed justification.

Response: MSPSS divided the sources of support into three sources. To refer to overall social support may not give sufficient information because each source may have a different impact at different stages of life. In particular, to children and adolescents, family support is important, while adults may find significant others such as doctors and bosses important as well[7]. Therefore, we have created four models to compare the effects of each source of social support. The models M1–M4 denote the mediators of overall support, family support, friends’ support, and significant others’ support respectively.

How do the authors define the direct/indirect influence of social support on the occurrence of depression in students with ADHD symptoms? Is this approach justified only for the studied group, or can it be considered a mechanism of general importance for the population?

Response: As far as we know, this can indicate only the association in a single plane due to its cross-sectional nature. We may define either a direct or indirect effect in a longitudinal study. We are confident only that this mechanism happens among first-year medical students. However, we conjecture that it might happen in other populations as well based on the significant findings in the pairing of ADHD symptoms and the pairing of perceived social support and depression.

Reviewer 2 Report

The authors are attributing depression symptoms to ADHD, yet, according to DSM criteria, to make a diagnosis of ADHD: * "The symptoms do not happen only during the course of schizophrenia or another psychotic disorder. The symptoms are not better explained by another mental disorder (e.g. Mood Disorder, Anxiety Disorder, Dissociative Disorder, or a Personality Disorder)." It is much more likely that Depression accounts for the ADH type symptoms of restlessness, inattentiveness, etc. than the other way around. There is also literature that inadequate sleep can result in ADH type symptoms and the rigors of medical school may well result in inadequate sleep. The paper goes into some detail on the statistical and analytic methods used for their models, for which I do not have the expertise to comment.

Author Response

(The authors gave the same response as above.)

Round 2

Reviewer 1 Report

I still believe that the chosen goal of the work is impossible to achieve. The first step should be to determine the scale of the ADHD problem among Taiwanese medical students. Otherwise, there can be no separation of depression from depression in the course of ADHD. The authors' answer did not solve the existing problem. I suggest completing this section based at least on the data collected in this study.

The authors cannot establish the relationship of social support with depression or anxiety and ADHD since they cannot confirm/indicate/investigate the prevalence of ADHD among Taiwanese medical students. The authors' answer did not solve the existing problem – as above.

It is unethical to examine young people with their consent and, in the case of an observed or even measured problem, no referral for consultation. After all, they are supposed to be doctors - maybe even future psychiatrists -able to help other patients. Commentary on the needs and possibilities should be included in the discussion as it justifies the purposefulness of similar research.

In Table 1, p-values should be provided to compare ADHD vs. non-ADHD symptoms to prove differences or lack thereof. This is the basis for the omission or the need to conduct analyzes in subgroups.

The definition of the direct/indirect influence of social support predicted by models should be supplemented

Author Response

Thank you again for your comments and concern. We highly appreciate that. We have made our best attempt to address on these concerns. Please see below our point-by-point responses.  We have added some sentences as suggested by the reviewer in our 2nd revised manuscript. The new added sentences are in green color.

I still believe that the chosen goal of the work is impossible to achieve. The first step should be to determine the scale of the ADHD problem among Taiwanese medical students. Otherwise, there can be no separation of depression from depression in the course of ADHD. The authors' answer did not solve the existing problem. I suggest completing this section based at least on the data collected in this study.

Response: We thank you for your suggestion.

We totally agree that the diagnoses of both ADHD and depressive disorder are essential.  This is important and really the first step to realize the magnitude of such problems in Thailand.  You are absolutely right that the diagnosis should have been established to clearly demonstrate the association as we planned. Unfortunately, we do not have robust evidence of the prevalence, and our research also did not support obtaining that data due to the measurements used. We realized that problem and have endeavored to make it clear that we were able to study the “ADHD and depressive symptoms”

However, as suggested by the reviewer, we have added some more statements to clarify this point in the introduction and limitation.

Although we cannot accurately estimate the magnitude of the ADHD problem, the fact that ADHD symptoms interfere with psychological wellbeing remains important and relevant. Research has revealed that 27.2% of medical students experience depression, 11% of whom experience suicidal ideation[2]; the authors would therefore like to focus on examining depression related to ADHD symptoms, even though the definite rate of prevalence of ADHD is unknown. 

Limitation:

Lastly, this study represents only the relationship of ADHD symptoms and depressive symptoms, without definite diagnosis of ADHD and depressive disorders; we cannot confidently conclude about the magnitude of the mediated effect. Moreover, a further study should be conducted to examine the scale of ADHD, comorbid depressive disorders and its relationship among adolescents as well as Thai medical students.

The authors cannot establish the relationship of social support with depression or anxiety and ADHD since they cannot confirm/indicate/investigate the prevalence of ADHD among Taiwanese medical students. The authors' answer did not solve the existing problem – as above.

Response: We thank you for your suggestion.

We agree with your point that we cannot confirm/indicate/investigate the prevalence of ADHD among Thai medical students as the study used only a screening tool. However, in terms of exploring the relationship between social support with depression, we believe that it can probably be achieved. Although the prevalence may influence the relationship among symptoms, our study does not claim the relationships occurred in the ADHD sample (because we did not establish definite ADHD status as the reviewer has mentioned). Therefore, we carefully emphasized on using the terms ‘symptoms’ to clarify to the reader our main research purpose.  

By the way, we think that you may misunderstand where our study was conducted. We conducted this study in Thailand, not Taiwan.

It is unethical to examine young people with their consent and, in the case of an observed or even measured problem, no referral for consultation. After all, they are supposed to be doctors - maybe even future psychiatrists -able to help other patients. Commentary on the needs and possibilities should be included in the discussion as it justifies the purposefulness of similar research.

Response: We appreciate your concern about the students’ wellbeing, so do we. In fact, we have many channels for the students who would like to receive consultation about their difficulties regarding their psychiatric symptoms as well as other challenges.  For research, we are allowed by the Ethics committee, to perform only those actions indicated in the proposal, PIS, and ICF. While we are concerned about the students’ wellbeing, the Ethics also is concerned about putting a stigma on the participant. The Ethics do not allow the researchers to inform the participants based on their screening scores (except for suicidality) according to confidentiality. The Ethics judgement is understandable as subjects would recognize how severe their symptoms are and could decide on their own whether to request consultation or not.   We have added this in discussion as detailed below.

Some concern should be raised here is as what action to take when some students reported high score on ADHD, depressive symptoms. No doubt these symptoms impact on the students’ wellbeing as well as their academic performance. The researcher is normally not allowed to provide direct interpretation according to the participant’s confidentiality (except for suicidality). However, this concern is mitigated as students are usually informed about consultation services when they are aware of having problems instigated by those questionnaires.  

In Table 1, p-values should be provided to compare ADHD vs. non-ADHD symptoms to prove differences or lack thereof. This is the basis for the omission or the need to conduct analyzes in subgroups.

Response. We have added p-values, Thank you.

The definition of the direct/indirect influence of social support predicted by models should be supplemented

Response. The direct/indirect effect was considered significant when the 95% confidence interval for the standardized estimated coefficient of the direct/indirect effect did not include zero.

We have added these sentences to the analysis plan.

Best regards,

NK, TW, and colleagues
